# AutoPDL: Automatic Prompt Optimization for LLM Agents

**Claudio Spiess**[1] **Mandana Vaziri**[2] **Louis Mandel**[2] **Martin Hirzel**[2]

[1]UC Davis
[2]IBM Research

**Abstract** The performance of large language models (LLMs) depends on how they are prompted, with choices spanning both the high-level prompting pattern (e.g., Zero-Shot, CoT, ReAct, ReWOO) and the specific prompt content (instructions and few-shot demonstrations). Manually tuning this combination is tedious, error-prone, and specific to a given LLM and task. Therefore, this paper proposes AutoPDL, an automated approach to discovering good LLM agent configurations. Our approach frames this as a structured AutoML problem over a combinatorial space of agentic and non-agentic prompting patterns and demonstrations, using successive halving to efficiently navigate this space. We introduce a library implementing common prompting patterns using the PDL prompt programming language. AutoPDL solutions are human-readable, editable, and executable PDL programs that use this library. This approach also enables source-to-source optimization, allowing human-in-the-loop refinement and reuse. Evaluations across three tasks and seven LLMs (ranging from 3B to 70B parameters) show consistent accuracy gains ($9.06 \pm 15.3$ percentage points), up to 68.9pp, and reveal that selected prompting strategies vary across models and tasks.

## 1 Introduction

Large language models (LLMs) and LLM-based agents excel at a variety of tasks, including question answering, math word problems, and programming. The performance of an LLM depends heavily on how it is prompted, and there are a variety of popular prompting patterns. These include zero-shot or few-shot (Brown et al. 2020) prompting with chain-of-thought (CoT) (Wei et al. 2022), zero-shot CoT (Kojima et al. 2022), as well as agentic patterns such as ReAct (Yao et al. 2023) or ReWOO (Xu et al. 2023). However, given a dataset $D_{\text{test}}$ with a loss function $\mathcal{L}$, *e.g.*, error rate, it is not clear which pattern will do best. Furthermore, besides the pattern $A$, performance also depends on the prompt $p$, including few-shot samples and instructions. The problem is thus to find a combination $A_p^*$ of a pattern along with an optimized prompt that minimizes $\mathcal{L}$. This is usually done via manual prompt engineering, but that is tedious and has to be repeated if a new LLM comes along. Therefore, this paper explores how to find $A_p^*$ using automated machine learning (AutoML). And for users to trust the result or to tweak it further, $A_p^*$ itself should be easy to read and edit.

Agent frameworks, such as CrewAI (Moura 2023) or AutoGen (Q. Wu et al. 2023), contain pre-built agent patterns with prompts optimized for proprietary frontier models and common tasks. Unfortunately, their prompts are deeply buried (Schluntz et al. 2024), making them hard to modify and adapt to non-frontier models or novel tasks. Moreover, the prompting pattern is fixed to a variation of ReAct, limiting flexibility in customizing the prompt structure. Prompt optimizers, such as DSPy (Khattab et al. 2024), optimize few-shot samples for in-context learning (ICL) and/or instructions in the prompt $p$. Unfortunately, they do not automatically select the prompting pattern $A$ and do not return human-readable code.

We formulate the problem of finding a good pattern and corresponding prompt by defining and then exploring a combined search space. We were inspired by the AutoML literature on combined search spaces of machine-learning algorithms and their hyperparameters (Thornton et al. 2013), except that (i) instead of discrete or continuous hyperparameters, we explore textual ICL samples, instructions, and prompting patterns; (ii) instead of classification or regression tasks, we tackle

generative tasks; and (iii) instead of model training or fine-tuning, we focus on in-context learning. We assume a dataset with a validation set $D_{\text{valid}}$, test set $D_{\text{test}}$, as well as an example bank $D_{\text{train}}$ for few-shot samples. $D_{\text{valid}}$ is used during the optimization process to evaluate the performance of a configuration, whereas $D_{\text{test}}$ is used once upon completion, to evaluate the performance of the best performing configuration. As usual, to avoid over-fitting, we assume these are disjoint from each other. The problem statement is to find $A_p^* = \underset{A_p \in \mathcal{A}_{\mathcal{P}}}{\text{argmin}} \, \mathcal{L}(A_p, D_{\text{valid}})$, where:

- $A \in \mathcal{A} = \{\text{Zero-Shot, CoT, ReWOO, ReAct}\}$ is the prompting pattern, and

- $p = \langle n, d_{\text{train}}, \text{instr} \rangle \in \mathcal{P}$ is the prompt, comprising a number $n \leq |D_{\text{train}}|$ of few-shot samples, the actual few-shot samples $d_{\text{train}} \in (D_{\text{train}})^n$, and an instruction $\text{instr} \in \mathcal{I}$. A concrete example of $\mathcal{P}$ is provided in Figure A1 and Figure A2.

To avoid getting stuck in local minima while saving compute and finding a solution with a low loss, we explore the search space $\mathcal{A}_{\mathcal{P}}$ using successive halving (Jamieson et al. 2016). To make the initial search space $\mathcal{A}_{\mathcal{P}}$ user-interpretable, and the final solution $A_p^*$ both human readable and executable, we express them in a YAML-based prompting language, PDL (Vaziri et al. 2024). PDL's structured format makes it easy to modify both the initial search space and the optimized program, and ensures the final solution remains directly executable. We introduce a library for PDL that implements each of the common prompting patterns in $\mathcal{A}$. The initial search space $\mathcal{A}_{\mathcal{P}}$ is a YAML file with various choice points for AutoML to decide. And the solution $A_p^*$ is a custom-tailored PDL program optimized for the given task, as given by the dataset and loss function. The developer can read or even tweak either or both as desired.

We evaluate our optimizer on three tasks (question answering, math, and programming), using seven LLMs sized between 3B and 70B parameters. We find that the optimizer often gives accuracy boosts in the 6–30% range, in some cases higher. Given the same task, different patterns $A \in \mathcal{A}$ do best for different models. Conversely, given the same model, different patterns do best for different tasks. Besides this variability in the chosen pattern $A$, our experiments also revealed variability in the optimized prompts $p = \langle n, d_{\text{train}}, \text{instr} \rangle$. We also found that when training data for a task is missing, data from a related but different dataset can help. Also, while most of our experiments use moderately-sized open models, we also show our optimized solutions can benefit frontier models.

This paper makes three primary contributions:

1. Jointly searching pattern and prompt: prior work in prompt optimization has not investigated searching *joint* search spaces, including agentic patterns.

2. No one size fits all: we find that different models sometimes have differing optimal prompt patterns for the *same* benchmark, suggesting that there is not one single optimal prompt pattern.

3. Source-to-source optimization: we propose the first source-to-source optimizer for LLM prompt programs, where both the initial search space and the final solution are prompt programs in the same language, making the final solution both human-readable and executable.

Overall, this paper shows how to apply AutoML to automatically discover agentic or non-agentic LLM prompts and patterns optimized for a given task. We make our AutoPDL implementation available at https://github.com/IBM/prompt-declaration-language, and release the reproduction package used for the experiments in this work to the community.

## 2 Background

This paper uses PDL (Vaziri et al. 2024) as a representation for exploring the search space of programs. PDL programs are declarative and combine human readability with ease of execution. They represent the composition of calls to LLMs and tools, abstracting away the plumbing necessary for

```
1  text:
2  - role: tools
3    text: ${ tools }
4  - "Out of 1400 participants, 400 passed the test. What percentage is that?\n"
5  - def: actions
6    model: replicate/ibm-granite/granite-3.1-8b-instruct
7    parser: json
8    spec: [{ name: str, arguments: { expr: str }}]
9  - if: ${ actions[0].name == "calc" }
10   then:
11     lang: python
12     code: result = ${ actions[0].arguments.expr }
```

Figure 1: Basic example of a PDL program.

such compositions. The output of the optimizer is also a PDL program, rather than simple textual prompts, so it is fully executable and could be further refined by a developer.

Figure 1 shows a simple PDL program that uses a tool to answer a query. PDL is based on the premise that interactions with an LLM are mainly for the purpose of generating data. So, it allows users to specify the shape of data to be generated in a declarative way (in YAML), and is agnostic of any programming language. The first line of Figure 1 specifies that we are creating some text. Next, the first block in the itemized list defines the tools prompt. Line 3 contains a use of variable *tools*, expressed as a Jinja expression. This variable is defined as the JSON Schema specification of a calculator tool (not shown in this figure, for the full program see Appendix 8.2). Line 4 is the user query prompt. We do not specify the role explicitly as *user* is the default role for prompts. Lines 5 through 8 show a model call. In PDL, the background context is accumulated implicitly, so the output of all blocks executed so far will be passed to the LLM as a list of input messages. The result of the model call is assigned to the variable *actions* (line 5). The model to be called is specified on line 6 (PDL is based on LiteLLM,[1] so this is a LiteLLM model id). Finally, lines 7 and 8 say that we parse the output of the model as JSON and type-check it according to the type on line 8. Furthermore, when the inferencing server supports it, model calls with a schema use constrained decoding (Willard et al. 2023), enforcing syntactically correct JSON.[2]

On line 9, an if-statement checks whether the output of the LLM asks for the calculator tool. If so, we use a Python code block to compute the requested tool call (lines 11 and 12). When we execute this program using the PDL interpreter, we obtain all the model inputs, followed by the model output, and finally the output of the tool call. PDL has a rich set of control structures to allow writing a variety of prompting patterns, as well as functions to support libraries. For instance, Figure 3 shows a function call on line 4. In this paper, we consider the problem of automatically tuning prompts and choosing prompting patterns for a given dataset. The following section explains our approach in further detail.

## 3 AutoPDL Approach

Figure 2 gives an overview of our approach. Referring to the numbers in the arrows:

(1) The input task is given by two disjoint datasets $D_{\text{train}}$ and $D_{\text{valid}}$ and a loss function $\mathcal{L}$. The datasets comprise $\langle x, y \rangle$ instances, where $x$ is a question, $y$ is the corresponding answer, and both are text strings. The loss function evaluates the quality of an answer $y$. (2) The search space specification $\mathcal{A}_{\mathcal{P}}$ is a YAML file with the optimization variables and their possible values, along with some hyperparameters. For example, `num_demonstrations: [0, 3, 5]` indicates that each candidate will have zero, three, or five ICL samples randomly drawn from $D_{\text{train}}$. In the case of zero demonstrations, this is equivalent to the zero-shot baseline. If zero is an option, we bias our candidate

---

[1] https://github.com/BerriAI/litellm

[2] PDL additionally makes use of the heuristic `json-repair` package.

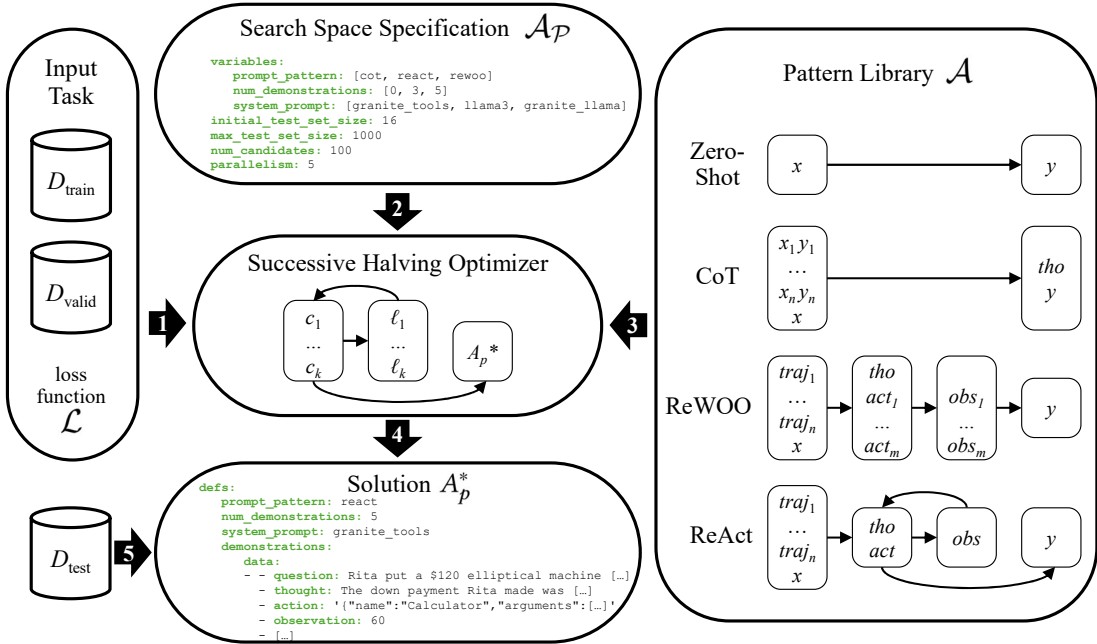

Figure 2: Overview of our approach.

sampling to always include one zero-shot candidate, just in case that baseline turns out to be the best-performing configuration.

(3) The pattern library consists of four PDL functions. Zero-shot is a baseline that simply prompts the LLM with $x$ and expects it to return $y$. CoT refers to chain-of-thought (Wei et al. 2022) with in-context learning (Brown et al. 2020): the input includes a few $x_i y_i$ pairs before the actual question $x$, and the output includes some reasoning thought *tho* before the actual answer $y$. ReWOO (Xu et al. 2023) refers to reasoning without observations. Here, the few-shot samples are trajectories $traj_i$. A trajectory $traj_i$ consists of steps for a particular example problem instance in $D_{\text{train}}$, *e.g.*, in the case of ReWOO, *tho* and *act* steps and their contents. In contrast to $x_i y_i$ pairs, $traj_i$ may contain many *tho*, *act* and, depending on pattern, *obs* steps, before reaching the solution $y_i$. The first LLM call in ReWOO generates one reasoning thought *tho* and multiple actions $act_i$. The PDL code executes each of the actions as a tool call to obtain the corresponding observations $obs_i$. A final model call generates the answer $y$ based on the observations. Finally, the ReAct pattern (Yao et al. 2023) starts with few-shot trajectory examples $traj_i$ (brief example visible under Solution $A_p^*$ in Figure 2) and the question $x$, and then enters a TAO (thought, action, observation) loop. In each loop iteration, the LLM generates *tho* and *act*, then the PDL code executes the action as a tool call, and feeds the tool output back as an observation *obs*. A special Finish action breaks out of the loop to return the answer $y$.

Once inputs (1), (2), and (3) are in place, the successive halving optimizer runs in a loop. It starts with a small subset $D_v \subset D_{\text{valid}}$ and many candidates $C = \{c_1, \ldots, c_k\} \subseteq \mathcal{A}_\mathcal{P}$ sampled from the search space. Each iteration uses $D_v$ to evaluate the corresponding losses $\ell_1, \ldots, \ell_k$. Then each iteration keeps the $\frac{k}{2}$ candidates with the smallest $\mathcal{L}$ while doubling the size of the validation subset $D_v$. See § 8.3 for the algorithm. (4) After the last iteration, the best remaining candidate is the solution $A_p^*$. This solution is a set of PDL definitions with concrete values for the optimization variables,

```
text:
  - include: ../../tools.pdl
  - include: ../../ReAct.pdl
  - call: ${ react }
    def: ANSWER
    args:
      task: "Question: ${ question }"
      model: ${ model }
      tools: ${ tools }
      trajectories: ${ demonstrations }
  - "\nThe answer is ${ ANSWER.answer }"
```

Figure 3: Basic example of PDL program using the ReAct pattern.

e.g., in Figure 2, `num_demonstrations:` 5 and `demonstrations:` ... a list of ReAct trajectories. (5) This program can be used on the test set $D_{\text{test}}$. For instance, Figure 3 shows a call to the ReAct function that passes `${demonstrations}` from $A_p^*$ as an argument.

## 4 Methodology

This section describes the datasets used, the tools available to agents, and our experimental setup, including how we construct agent trajectories that demonstrate tool use for each dataset.

### 4.1 Datasets

We selected datasets that are widely used in the literature, span diverse tools and domains, and are representative of tool categories frequently studied in prior work (*e.g.*, calculator, search, code execution). In our experiments, each dataset has three disjoint splits: $D_{\text{train}}$ to sample few-shot samples from, $D_{\text{valid}}$ to evaluate candidates during optimization, and $D_{\text{test}}$ to evaluate the final chosen solution upon completion of optimization.

**GSM8K.** The Grade School Math (Cobbe et al. 2021) dataset consists of 8,792 grade school math problems. We sample 1,024 problems without replacement from the train set to use as our $D_{\text{valid}}$ set, and 1,024 from the test set (consisting of 1,319 samples) to use as our $D_{\text{test}}$. This leaves a $D_{\text{train}}$ of 6,449 problems. Each problem consists of a word problem $x$ such as *"What is fifteen more than a quarter of 48?"*, a sequence *tho* of reasoning steps, and finally a plain numeric answer $y$ following a special delimiter. For the CoT prompt pattern, we include these reasoning steps directly in the demonstrations. We use a regular expression to extract the numerical answer from the model solution, and define a correct solution as an exact match to the ground truth answer.

**GSM-Hard.** Gao et al. (2023) introduce a derivative of GSM8K with variables randomly changed to large numbers, with 1,319 samples. Unfortunately, GSM-Hard had 132 samples where the ground truth was incorrect, and hence we excluded those samples. We split the single set into equally sized $D_{\text{valid}}$ and $D_{\text{test}}$ ($n = 594$), and use the GSM8K training set (6,449 samples) described above for $D_{\text{train}}$ (*cross transfer*). We use the same correctness criterion as for GSM8K.

**FEVER.** The Fact Extraction and VERification dataset (Thorne et al. 2018) is a question-answering dataset structured around fact-checking. The original dataset contained 185,445 claims that are true, false, or unverifiable, and associated with human annotated supporting, refuting, or neutral sentences and their Wikipedia article of origin. We follow the widely used derivative of this benchmark in BIG-bench (Srivastava et al. 2023), which reformulates it into a true-or-false task by removing unverifiable claims. We sample 1,024 claims from the train set as $D_{\text{valid}}$ and 1,024 from the test set as $D_{\text{test}}$. This leaves a $D_{\text{train}}$ of 5,696 claims. BIG-bench also does not include the supporting or refuting sentences, which we recover by joining on the original dataset, for use *e.g.*, as CoT demonstrations. To assess correctness, we check for the presence of "true" and "false" in the final line of the model response. If neither or both are present, we deem the response as incorrect, and otherwise the correctness is a direct match to the ground truth "true" or "false".

**MBPP+.** MBPP (Austin et al. 2021) is a dataset of 974 mostly basic Python problems, with each example consisting of a natural language problem specification $x$ for a self-contained Python function $y$, along with a single test case. Each problem has an extended set of test cases used for evaluation, which are not shown to the model. Liu et al. (2023) found that MBPP test cases are incomplete, allowing proposed solutions to pass as correct, despite not matching the problem specification. Therefore, our experiments are based on MBPP+, which contains a subset of the problems, but a more complete set of test cases for each problem. We use these test cases to assess the correctness of the proposed solutions. We use the 374 examples from the original MBPP dataset as $D_{\text{train}}$, and split MBPP+ into $D_{\text{test}}$ (224 samples) and $D_{\text{valid}}$ (39 samples) based on which split they were in

*MBPP.* The discrepancy in number of MBPP+ samples is due to the exclusion of samples found in the MBPP train set, to avoid data contamination. Our ReAct implementation follows Wang et al. (2024). We do not implement ReWOO as it is not reactive, *i.e.*, cannot include execution feedback.

## 4.2 Tools

Our prompt library represents actions, *i.e.*, tool calls, following the JSON tool calling schema (Abdelaziz et al. 2024). An action is represented as a JSON object with `name` and `arguments` mapping. For example, `{"action": "Calc", "arguments": {"expr": "48/4"}}` represents a call to a calculator with the expression to evaluate. The PDL functions implementing the patterns (see Figure 2) accept a list of tool definitions as an argument. Each element of that list is itself a `PDL` function. As both the agents and tools are implemented in PDL, the set of tools for a given task could itself be made a search space dimension, which we leave to future work.

**Calculator.** For math datasets, we give the agentic approaches access to the *Calc* tool. This tool evaluates a cleaned (*e.g.*, replacement of `^` with `**`) expression with SymPy, returning the result. In case of error, the function returns a warning that the expression was invalid, which may help the agent recover from invalid input.

**Search.** For fact verification, we provide access to the *Search* tool, which returns the summary of the first Wikipedia search result for the query. If no results are found, a hint to try again is returned, or if the title is too ambiguous, a list of possible disambiguations.

**Execute.** We implement a programming agent for the code generation task, which can execute arbitrary code surrounded in XML-style `<execute>` tags. This tool executes the code in a Python shell, which returns the result of the final expression. This allows the agent to test its proposed solution against the given test case before submitting its solution.

**Finish.** The most basic action is the *Finish* action, or `<solution>` tag for the coding agent. This ends the agent's trajectory and results in the agent returning the value as the solution.

## 4.3 Experimental Setup

We evaluate the efficacy of our approach by running an optimization process to completion for each model & dataset pair, and subsequently comparing the task accuracy. As a baseline, we evaluate each model in a zero-shot setting. This setting reflects the minimal effort approach by a user or developer, where they do not include any demonstrations with their query or task. As no model we investigate was specifically trained to create agentic trajectories in a zero-shot setting, it is not feasible to create a zero-shot setting under ReAct or ReWOO.

For the optimization process, we used an initial candidate set $\mathcal{C}$ of 100 candidate prompt configurations $c_i = \langle A, p \rangle \in \mathcal{C}$ per experiment. We fixed the size of the initial validation subset $d_{\text{valid}} \subseteq D_{\text{valid}}$ to 16. We define $\mathcal{L}(c_i, d_{\text{valid}}) = -\text{Accuracy}(c_i, d_{\text{valid}})$ for a given candidate $c_i$, where accuracy is the fraction in $d_{\text{valid}}$ of correctly solved problems by $c_i$ as per the dataset definition of correctness. For each candidate $c_i$, $p = \langle n, d_{\text{train}}, \text{instr} \rangle \in \mathcal{P}$ where $n$ few-shot samples or trajectories $d_{\text{train}} \in (D_{\text{train}})^n$ are randomly sampled with replacement. The number of possible values for $\mathcal{P}$ is combinatorially large and depends on the dataset $D_{\text{train}}$ used. Finally, upon completion of an optimization process, the optimal candidate is evaluated on $D_{test}$.

**Constructing Trajectories.** As we are also optimizing over agentic trajectories, we also need a set of trajectories to sample few-shot samples from. To achieve this, we create a basic agentic trajectory $traj_i$ for each training example $\langle x_i, y_i \rangle$, following a rule-based transformation. We design and apply a template to each dataset, which is relatively simple and easy to implement for other datasets (details are provided in § 8.5). Prior work has introduced approaches to bootstrapping trajectories *e.g.*, in software engineering (Pan et al. 2024), tool use demonstrations (Li et al. 2025),

and reasoning paths (Zelikman et al. 2022), which could be applied to this problem. While we acknowledge the shortcomings of manual template construction, we argue this approach has two strengths: it is generalizable in the sense that templates can be mixed and matched, and that the trajectories are directly based on the datasets used. Additionally, we wanted to work with commonly used datasets that cover a variety of tools and domains, rather than emerging datasets containing trajectories or tool use demonstrations.

**Models**. We aim to study models of various abilities, *e.g.*, natural language or code, various creators, and various sizes, ranging from single digit billions of parameters up to the edge of feasibility on consumer hardware. We include seven models available on inference service IBM watsonx[3] in our study. We select three generalist natural language instruction models, LLAMA 3.1 8B, 3.2 3B, and 3.3 70B from the open-source and widely studied LLaMa family (Dubey et al. 2024). We further select three models from Mishra et al. (2024), which predate Dubey et al. (2024) by approximately 3 months. We select GRANITE 13B INSTRUCT V2 as a generalist model, and GRANITE 20B and 34B CODE INSTRUCT as code models. We select GRANITE 3.1 8B as an additional generalist model (Granite Team 2024). All of our experiments use greedy decoding, *i.e.*, no sampling, to limit the impact of hyperparameter choice. The number of models we evaluate is limited by cost in $/token terms and execution time. By studying various models, we demonstrate the generalizability of our approach.

**Alternative Setups**. We evaluate two alternative experimental setups. First, to investigate low-resource scenarios, we examine whether performance on one dataset can be improved by using demonstrations $D_{\text{train}}$ from a similar dataset, while optimizing w.r.t. $D_{\text{valid}}$. For this experiment, we investigate whether performance on GSM-Hard can be improved by using demonstrations from *GSM8K*, while optimizing w.r.t. GSM-Hard $D_{\text{valid}}$. Second, to explore saving optimization costs, we assess whether optimized prompt programs of one model can transfer well to a frontier model. The intuition is that while that might not be the best program for the frontier model, it might at least improve somewhat over the baseline. To this end, we evaluate the optimized PDL programs of LLAMA 3.1 70B on OpenAI's `gpt-4o-mini-2024-07-18`, for each dataset.

## 5 Results

This section describes the results of our empirical study to evaluate our AutoPDL approach and answer the following research questions:

> **RQ 1**: To what extent does AutoPDL improve accuracy, and how much does the best solution vary by task and model?

RQ1 asks to what degree our AutoPDL approach can improve model performance over their zero-shot baseline across a variety of commonly used benchmarks. We also seek to identify trends, if any, in optimal configurations, *e.g.*, whether more few-shots is always better, or whether certain prompt patterns are particularly suited to certain problem domains.

> **RQ 2**: Can AutoPDL make up for a missing few-shot example bank for a given task by reusing the example bank from a similar task?

RQ2 investigates whether optimizing on one dataset using demonstrations from another, related, dataset can result in higher performance than using no demonstrations (zero-shot). This RQ addresses a low-resource scenario in which a limited pool of demonstrations exists in one dataset, but a dataset from a similar domain has a large pool.

> **RQ 3**: Do solutions found by AutoPDL improve performance on frontier models, even when optimized for open-source models?

---

[3] https://www.ibm.com/watsonx

It can be expensive to run optimization against commercial frontier-model APIs. RQ3 assesses whether optimized prompt programs are transferable to different (and likely stronger) models than those they were optimized with.

Table 1 reports the results of our optimization and evaluation procedure. We performed three complete optimization runs for GSM8K and MBPP+, and report mean accuracy and standard deviation in percentage points (*i.e.*, absolute, *not* relative, uncertainty). We did not complete multiple runs for all model-dataset pairs due to time and resource constraints. Where multiple runs were performed, we report the pattern of the highest scoring run. Additionally, we note minor variance in certain zero-shot settings due to API non-determinism. Across models and datasets, we generally find some improvement over the zero-shot baseline with few-shot chain-of-thought, or agentic patterns ReAct or ReWOO.

Table 1: Model accuracies across datasets for baseline (zero-shot) and optimized versions.

| Dataset | Model | Accuracy | | | Pattern | Runtime |
|---|---|---|---|---|---|---|
| | | Zero-Shot | Optimized | Delta | | |
| FEVER | Granite 3.1 8B | 78.3 % | 79.0 % | +0.7pp | ReWOO (5 shot) | 08:55 |
| | Granite 13B Instruct V2 | 6.5 % | 75.4 % | +68.9pp | ReWOO (3 shot) | 08:12 |
| | Granite 20B Code | 39.7 % | 64.2 % | +24.5pp | CoT (3 shot) | 05:06 |
| | Granite 34B Code | 56.4 % | 65.6 % | +9.2pp | CoT (3 shot) | 03:47 |
| | LLaMA 3.1 8B | 68.5 % | 78.0 % | +9.5pp | CoT (3 shot) | 05:24 |
| | LLaMA 3.2 3B | 38.0 % | 66.9 % | +28.9pp | ReWOO (5 shot) | 09:08 |
| | LLaMA 3.3 70B | 67.6 % | 77.5 % | +9.9pp | ReWOO (5 shot) | 09:32 |
| GSM8K | Granite 3.1 8B | 74.2 % | $(74.2 \pm 0.6)$ % | +0.0pp | Zero-Shot (Baseline) | 08:56 |
| | Granite 13B Instruct V2 | 23.0 % | $(30.9 \pm 1.0)$ % | +7.9pp | CoT (3 shot) | 09:20 |
| | Granite 20B Code | 68.7 % | $(68.7 \pm 0.1)$ % | +0.0pp | Zero-Shot (Baseline) | 09:27 |
| | Granite 34B Code | 72.1 % | $(72.1 \pm 0.1)$ % | +0.0pp | Zero-Shot (Baseline) | 08:52 |
| | LLaMA 3.1 8B | 78.4 % | $(85.3 \pm 0.6)$ % | +6.9pp | CoT (5 shot) | 08:48 |
| | LLaMA 3.2 3B | 71.8 % | $(75.3 \pm 0.4)$ % | +3.5pp | CoT (3 shot) | 16:36 |
| | LLaMA 3.3 70B | 85.5 % | $(95.4 \pm 0.2)$ % | +9.9pp | CoT (3 shot) | 07:50 |
| MBPP+ | Granite 3.1 8B | 62.9 % | $(62.9 \pm 0.0)$ % | +0.0pp | Zero-Shot (Baseline) | 02:14 |
| | Granite 13B Instruct V2 | 10.7 % | $(19.2 \pm 1.2)$ % | +8.5pp | ReAct (5 shot) | 04:02 |
| | Granite 20B Code | 51.8 % | $(51.8 \pm 0.4)$ % | +0.0pp | Zero-Shot (Baseline) | 03:43 |
| | Granite 34B Code | 48.7 % | $(61.3 \pm 1.0)$ % | +12.6pp | ReAct (3 shot) | 04:54 |
| | LLaMA 3.1 8B | 61.2 % | $(62.8 \pm 4.0)$ % | +1.6pp | ReAct (5 shot) | 01:45 |
| | LLaMA 3.2 3B | 58.0 % | $(58.0 \pm 0.4)$ % | +0.0pp | Zero-Shot (Baseline) | 02:01 |
| | LLaMA 3.3 70B | 71.4 % | $(71.4 \pm 0.0)$ % | +0.0pp | Zero-Shot (Baseline) | 02:27 |

**FEVER.** We observed the minimum improvement in GRANITE 3.1 8B, with a 0.7 percentage point (pp) improvement, and a maximal improvement of 68.9pp for GRANITE 13B INSTRUCT V2. In terms of prompt pattern, CoT and ReWOO are equally represented. ReAct was not the optimal for any of the models. Interestingly, the largest model (LLaMA 3.3 70B) benefited by 9.9pp from 5-shot CoT. FEVER runtimes are generally higher than the other benchmarks, likely due to the large number of tokens involved by including Wikipedia content.

**GSM8K.** The highest improvement recorded (9.9pp) was for LLaMA 3.3 70B using 3-shot CoT, while the minimum improvement of 3.5pp was in LLaMA 3.2 3B using 3-shot CoT. ReAct and ReWOO were not the optimal for any model. For GRANITE 3.1 8B, GRANITE 20B CODE, and GRANITE 34B CODE, no improvement over the zero-shot baseline was identified. This was somewhat surprising, as generally including even some few-shot samples improves performance in LLMs.

**MBPP+.** Several models benefited from execution feedback, as 3 out of 7 had ReAct as the optimal prompt pattern (ReWOO was excluded as described in § 4.3). The greatest improvement of

12.6pp was in GRANITE 34B CODE, and 8.5pp in GRANITE 13B INSTRUCT V2, likely due to its poor programming performance as a generalist, non-code model. In contrast, the smaller LLaMa 3.1 8B model had high zero-shot performance of 61.2%, yet still improved by up to 6.2pp (1.6pp on average) with ReAct. No improvement was observed for GRANITE 3.1 8B, GRANITE 20B CODE, or the other LLaMa models.

Table 2: Model accuracies on GSM-Hard for cross optimization experiment.

| Dataset | Model | Accuracy | | | Pattern | Runtime |
|---|---|---|---|---|---|---|
| | | Zero-Shot | Optimized | Delta | | |
| | Granite 3.1 8B | 44.0 % | 44.0 % | +0.0pp | Zero-Shot (Baseline) | 04:57 |
| | Granite 13B Instruct V2 | 4.7 % | 5.6 % | +0.8pp | CoT (3 shot) | 03:30 |
| | Granite 20B Code | 28.8 % | 28.8 % | +0.0pp | Zero-Shot (Baseline) | 08:26 |
| GSM-Hard | Granite 34B Code | 27.9 % | 30.0 % | +2.0pp | ReWOO (5 shot) | 05:49 |
| | LLaMA 3.1 8B | 31.6 % | 32.3 % | +0.7pp | ReWOO (5 shot) | 04:44 |
| | LLaMA 3.2 3B | 26.3 % | 27.4 % | +1.2pp | CoT (5 shot) | 16:50 |
| | LLaMA 3.3 70B | 47.3 % | 53.4 % | +6.1pp | CoT (5 shot) | 11:07 |

Table 3: Model accuracy for GPT-4o-mini cross experiment results

| Dataset | Model | Accuracy | | | Pattern |
|---|---|---|---|---|---|
| | | Zero-Shot | Optimized | Delta | |
| FEVER | GPT-4o-mini | 83.7 % | 87.7 % | +4.0pp | CoT (3 shot) |
| GSM-Hard | GPT-4o-mini | 45.6 % | 54.9 % | +9.3pp | ReAct (5 shot, Granite LLaMa) |
| GSM8K | GPT-4o-mini | 77.8 % | 90.9 % | +13.1pp | CoT (5 shot) |
| MBPP+ | GPT-4o-mini | 72.3 % | 72.3 % | +0.0pp | Zero-Shot (Baseline) |

**Missing Few-Shot Example Bank**. We optimized the PDL program for GSM-Hard, using GSM8K demonstrations, and report results in Table 2. We found that in most cases, GSM8K demonstrations were at least not harmful for models on GSM-Hard, with up to 6.1pp improvement for LLaMa 3.3 70B using 5-shot CoT.

**Commercial Frontier Model**. To assess whether performance gains in one model can be achieved in another, we evaluate the optimized PDL programs of LLaMa 3.1 70B on OpenAI's `gpt-4o-mini-2024-07-18` and report results in Table 3 (we did not use LLaMa 3.3 70B here because we did this experiment earlier and did not have the time and resources to repeat it for the final version of this paper). For all dataset/prompt pattern pairs that resulted in improvement for LLaMa 3.1 70B, we found a surprising improvement in GPT4o-MINI of at least 4pp on FEVER using 3-shot CoT, 9.3pp on GSM-Hard (using GSM8K demonstrations) with 5-shot ReAct (Granite LLaMa instructions), and up to 13.1pp on GSM8K using 5-shot CoT. This suggests that optimizing for an open-source model can also benefit a closed-source model.

## 6  Related Work

The closest related work is on **prompt optimization**. APE starts with an LLM-generated set of candidate prompts, then performs rejection sampling based on evaluation on a subset of data (Zhou et al. 2023). ZOPO incorporates a Neural Tangent Kernel-based derived Gaussian process into standard zeroth-order optimization for an efficient search of a locally-optimal instruction (Hu et al. 2024). Unlike our approach, neither APE nor ZOPO optimize few-shot samples. Aviary can also jointly optimize over prompt pattern, instruction, and few-shot examples (Narayanan et al. 2024). However, it would require the definition of a custom operator. CEDAR uses a demonstration pool, from which it retrieves few-shot examples at query time (Nashid et al. 2023). Unlike our approach, these few-shot samples are retrieved on a per-inference basis, not optimized ahead-of-time.

EASE leverages embeddings to represent few-shot examples, and uses a neural bandit algorithm to find an ordered set that performs well for test queries from a given task (Z. Wu et al. 2024). An extension of their approach jointly optimizes demonstrations and instructions. However, the approach requires both an additional embedding model, and the training of a new model to predict validation scores from embeddings. Unlike our approach, EASE does not optimize over agentic patterns.

DSPy optimizes instructions and few-shot samples for a chain of LLM calls (Khattab et al. 2024) (not just a single call like APE or CEDAR). Also, DSPy takes away control over the exact prompt from the programmer, which our approach preserves. Similarly to DSPy, TextGrad also optimizes a chain of LLM calls, by using LLMs to back-propagate modifications to instructions in prompts (Yuksekgonul et al. 2025). However, unlike our approach, neither of these optimize agentic patterns.

BPO trains a sequence-to-sequence model on prompts augmented by an LLM incorporating human preferences, producing a model that improves given input prompts (Cheng et al. 2024). APOHF introduce a strategy to select a pair of prompts to query the user for preference feedback, which they use to optimize LLM-generated instructions on a validation set (Lin et al. 2024). However, neither of these approaches explicitly optimize demonstrations or agentic patterns. EvoAgent optimizes the instructions of a population of agents via crossover, mutation, and selection (Yuan et al. 2024). It then forms an ensemble from the final, fittest, population. GPTSwarm represents each agent as a graph, then freezes intra-agent edges and optimizes the placement of additional inter-agent edges (Zhuge et al. 2024). Unlike our approach, neither EvoAgent nor GPTSwarm optimize the agentic pattern inside individual agents, nor do they optimize few-shot samples.

Another closely related field of study is **AutoML**. Auto-sklearn (Feurer et al. 2015) used Bayesian optimization to jointly perform both algorithm selection and hyperparameters of a scikit-learn pipeline (Buitinck et al. 2013). While different, we see some analogy between algorithms and agentic patterns, and between hyperparameters and few-shot samples. DAUB first evaluates many candidate models on a small amount of data, then successively reduces candidates and increases data to ultimately pick a strong model (Sabharwal et al. 2016). The successive-halving algorithm takes a similar approach (Jamieson et al. 2016). Our approach is inspired by the same incremental data allocation idea. While both randomized search and Bayesian optimization are popular in AutoML, there are also more intricate approaches. For instance, TPOT uses genetic algorithms (Olson et al. 2016), and AlphaD3M uses Monte-Carlo tree search (Drori et al. 2018). We chose to start with a simpler technique that depends less on a well-behaved optimization space. That said, exploring more advanced AutoML optimizers could be fruitful future work for AutoPDL. Lale (Baudart et al. 2021) treats AutoML as a source-to-source optimization, similar to this paper, but unlike AutoPDL, it has not been used to optimize agentic patterns or prompts.

## 7 Conclusion

We present our AutoPDL approach for jointly optimizing prompting patterns and textual prompts for large language models, addressing the challenges associated with manual prompt engineering. By formulating the optimization as a discrete search over both agentic and non-agentic patterns, combined with instructions and few-shot samples, we leveraged successive halving to efficiently navigate this search space. Our evaluation across various datasets (FEVER, GSM8K, GSM-Hard, and MBPP+) and multiple models (from the LLaMA, Granite, and GPT families) demonstrates substantial accuracy improvements, up to 68.9 percentage points, and affirms that no single prompting strategy universally outperforms others across tasks and models. Additionally, generating code in a YAML-based prompt programming language (PDL) makes it executable, easy to modify, and readable by humans, supporting practical adoption and adaptation.

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

## 8 Supplemental Material

### 8.1 Concrete Prompt Example

```
1  defs:
2    prompt_pattern: cot
3    num_demonstrations: 2
4    demonstrations:
5      data:
6      - question: Tricia is a third of Amilia's age and Amilia is a quarter of Yorick's age. Yorick is twice
7          Eugene's age and Khloe is a third of Eugene's age. Rupert is 10 years older than Khloe but 2 years
8          younger than Vincent who is 22 years old. How old, in years, is Tricia?
9        reasoning: |-
10         Rupert is younger than Vincent by 2 years, so he is 22 years old - 2 years = <<22-2=20>>20 years old.
11         Khloe is 10 years younger than Rupert so she is 20 years old - 10 years = 10 years old.
12         Eugene is 3 times older than Khloe so he is 10 years old * 3 = <<10*3=30>>30 years old.
13         Yorick is twice Eugene's age so he is 30 years old * 2 = <<30*2=60>>60 years old.
14         Amilia is a quarter of Yorick's age so she is 60 years old / 4 = <<60/4=15>>15 years old.
15         Tricia is a third of Amilia's age so she is 15 years old / 3 = <<15/3=5>>5 years old.
16       answer: '5'
17     - question: Emmalyn decided to paint fences in her neighborhood for twenty cents per meter. If there were
18         50 fences in the neighborhood that she had to paint and each fence was 500 meters long, calculate
19         the total amount she earned from painting the fences.
20       reasoning: |-
21         The total length for the fifty fences is 50*500 = <<50*500=25000>>25000 meters.
22         If Emmalyn charged twenty cents to paint a meter of a fence, the total income she got from painting
23         the fences is $0.20*25000 =$5000
24       answer: '5000'
25   instruction: Answer the questions to the best of your abilities.
26 text:
27   - include: CoT.pdl
28   - "${ instruction }\n\n"
29   - call: ${ chain_of_thought }
30     args:
31       examples: ${ demonstrations }
32       question: ${ question }
33       model: ${ model }
```

Figure A1: Basic example of a concrete prompt configuration in PDL, using CoT pattern and two demonstrations.

```
1  Answer the questions to the best of your abilities.
2
3  Question: Tricia is a third of Amilia's age and Amilia is a quarter of Yorick's age. Yorick is twice Eugene's
4  age and Khloe is a third of Eugene's age. Rupert is 10 years older than Khloe but 2 years
5  younger than Vincent
6  who is 22 years old. How old, in years, is Tricia?
7  Answer: Let's think step by step. Rupert is younger than Vincent by 2 years, so he is 22 years
8  old - 2 years = <<22-2=20>>20 years old.
9  Khloe is 10 years younger than Rupert so she is 20 years old - 10 years = 10 years old.
10 Eugene is 3 times older than Khloe so he is 10 years old * 3 = <<10*3=30>>30 years old.
11 Yorick is twice Eugene's age so he is 30 years old * 2 = <<30*2=60>>60 years old.
12 Amilia is a quarter of Yorick's age so she is 60 years old / 4 = <<60/4=15>>15 years old.
13 Tricia is a third of Amilia's age so she is 15 years old / 3 = <<15/3=5>>5 years old.
14 The answer is 5
15
16 Question: Emmalyn decided to paint fences in her neighborhood for twenty cents per meter. If there were 50
17 fences in the neighborhood that she had to paint and each fence was 500 meters long, calculate the total
18 amount she earned from painting the fences.
19 Answer: Let's think step by step. The total length for the fifty fences is 50*500 = <<50*500=25000>>25000 meters.
20 If Emmalyn charged twenty cents to paint a meter of a fence, the total income she got from painting the
21 fences is $0.20*25000 =$5000
22 The answer is 5000
23
24 Question: ${ question }
25 Answer: Let's think step by step.
```

Figure A2: Corresponding rendered prompt configuration for Figure A1. With the exception of the non-rendered ${ question } variable, this is the input to the language model.

See Figure A1 and Figure A2.

## 8.2 Tool Calling Code

```
1  description: tool use
2  defs:
3    tools:
4      data:
5      - name: calc
6        description: Calculator function
7        arguments:
8          expr:
9            type: string
10            description: Arithmetic expression to calculate
11 text:
12 - role: system
13   text: You are Granite, developed by IBM. You are a helpful AI assistant
14     with access to the following tools. When a tool is required to answer
15     the user's query, respond with <|tool_call|> followed by a JSON list of
16     tools used. If a tool does not exist in the provided list of tools,
17     notify the user that you do not have the ability to fulfill the request.
18   contribute: [context]
19 - role: tools
20   text: ${ tools }
21   contribute: [context]
22 - "Out of 1400 participants, 400 passed the test. What percentage is that?\n"
23 - def: actions
24   model: replicate/ibm-granite/granite-3.1-8b-instruct
25   parser: json
26   spec: [{ name: str, arguments: { expr: str }}]
27 - "\n"
28 - if: ${ actions[0].name == "calc" }
29   then:
30     lang: python
31     code: result = ${ actions[0].arguments.expr }
```

Figure A3: Basic example of a PDL program.

See Figure A3.

## 8.3 Optimization

---

**Algorithm 1** Successive Halving for PDL Optimization

---

**Require:** Program candidate set $\mathcal{C}$, validation dataset $D_{\text{valid}}$, initial validation subset size $v_{\text{min}}$, maximum validation subset size $v_{\text{max}}$

1: $v \leftarrow v_{\text{min}}$
2: **while** $|\mathcal{C}| > 1$ **do**
3:     $d_{\text{valid}} \leftarrow$ first $v$ elements of $D_{\text{valid}}$ s.t. $d_{\text{valid}} \subseteq D_{\text{valid}}$ and $|d_{\text{valid}}| = v$
4:     **for** each candidate $c_i \in \mathcal{C}$ **do**
5:         Compute loss $\ell_i \leftarrow \mathcal{L}(c_i, d_{\text{valid}})$
6:     **end for**
7:     $\mathcal{C} \leftarrow$ top $\lfloor |\mathcal{C}|/2 \rfloor$ candidates with lowest loss
8:     $v \leftarrow \min(v_{\text{max}}, 2 \cdot v)$
9: **end while**
10: **return** Candidate in $\mathcal{C}$ with lowest loss

---

Figure A4: Illustration of the Successive Halving algorithm used to optimize the PDL program by pruning poor candidates on progressively larger validation subsets.

[Figure A4](#) describes our optimization algorithm, based on successive halving ([Jamieson et al. 2016](#)). The algorithm accepts a candidate set sampled from possible configurations and demonstrations, a validation dataset to optimize against, an initial validation subset size, and a maximum validation subset size. AutoPDL allows the user to specify these options in a YAML configuration file, and ultimately saves its result as a PDL program. This source-to-source transformation enables the user to modify both the search space and the resulting optimized PDL program, allowing further modification and execution.

## 8.4 Search Space

The search space is the Cartesian product of the following discrete variables, each taking one value per candidate:

(1) $A \in \mathcal{A} = $ {Zero-Shot, CoT, ReWOO, ReAct}, *i.e.*, the overall prompting pattern to apply.

(2) **Number of demonstrations** $n \in \{0,\ 3,\ 5\}$. We selected these options as a representative sweep across no supervision, moderate few-shot use, and an upper-end case (in terms of token window). We limited the search space to three options to avoid combinatorial explosion and limit experiment cost.

(3) If $A = $ ReAct, **System prompt** $\in$ {Granite Tools, LLaMa 3, Granite LLaMa}. As the system prompt instructs the model how to format tool calls, it only has an effect on benchmarks with JSON tool calling (FEVER, GSM8K, and GSM-Hard) for candidates with the ReAct prompt pattern. We note that only for MBPP+, ReWOO is not included as a prompt pattern, and that we always include two trajectories displaying iterative refinement, *i.e.*, an example of a solution failing the example test case, followed by a passing solution, in line with [Wang et al. (2024)](#). This effectively increases the number of trajectories to $|traj| + 2$.

## 8.5 Agent Trajectory Construction

To optimize over agentic patterns and trajectories, we require a set of example trajectories to use during optimization. For this purpose, we create a basic agentic trajectory $traj_i$ for each training example $\langle x_i, y_i \rangle$, following a rule-based transformation outlined below.

**GSM8K.** To demonstrate tool use in ReAct, we instead derive a trajectory *traj* as follows. We exploit the fact that there is at most one expression per reasoning step, by iterating through the steps. At each step, we append a 'thought' to the trajectory, consisting of the text leading up to the math expression, concatenated with a reflection 'I need to calculate'. We append a calculator tool call with the expression, and an 'observation', *i.e.*, the result of the expression. Finally, we append a thought 'The answer is …', containing the ground truth answer, followed by the *finish* action with the answer. We follow the same procedure to create ReWOO trajectories, except we use slightly different wording, *e.g.*, 'Calculate xyz' in place of 'I need to calculate xyz', and omit the final thought and action. Additionally, we use string substitution to replace any assumed expression results in the trajectory with the corresponding variable.

**FEVER.** To produce agent trajectories, we iterate over each article associated with a claim, append a thought 'I need to search for …', followed by the action, an observation containing the article summary, and finally a thought containing all the relevant sentences associated with that article for that claim, which we repeat for each article associated with a claim. This procedure is not ideal as there is no inherent order to the articles or sentences, even though there may be a natural ordering following the annotator's Wikipedia navigation. Finally, we append a thought 'The claim is true/false' and the finish action, both with the ground truth answer. For chain-of-thought, we perform the same procedure except we only include the concatenated evidence sentences, as there is no tool use.

**MBPP+.** To generate sample agent trajectories from the training set, we follow the agent pattern (without feedback) in-context examples by Wang et al. (2024), which consists of the problem $x$, a thought such as "The intersection is elements that are in both lists", an execute action that contains proposed code *and* an assertion calling the proposed method with the test case input from the prompt and comparing its output. This is then followed by an observation containing the execution result, *i.e.*, either "[Executed Successfully with No Output]" or a stack traceback. This allows the agent to iterate on solutions (up to five times in our implementation). We use the full MBPP train set of 374 problems as $D_{\text{train}}$, and split the MBPP+ dataset into $D_{\text{valid}}$ and $D_{\text{test}}$ based on problem id membership in MBPP, leaving 39 and 224 validation and test problems respectively.

To generate synthetic trajectories from the training set, we start with the natural language specification and single test case (the prompt), append the thought "I should run a solution on the test case before proposing a solution.", followed by the ground truth solution and substitute in the prompt test case following the pattern [solution]res = ...; assert res == ..., "Expected ... but got ".format(res). Subsequently, we append the observation "[Executed Successfully with No Output]", the thought "There is no more AssertionError. I can now submit the solution.", and finally the solution action with the ground truth solution. This naive approach allows us to provide demonstration trajectories, albeit simplistic ones that assume the first solution is correct. Sampling a reflection or thought from a strong model may be beneficial (Li et al. 2025), but we restrict our trajectories to rule based transformations. As ReWOO is not reactive, *i.e.*, without execution feedback, it does not make sense for MBPP. Hence, we exclude it from our experiments.

## 8.6 Results Plot

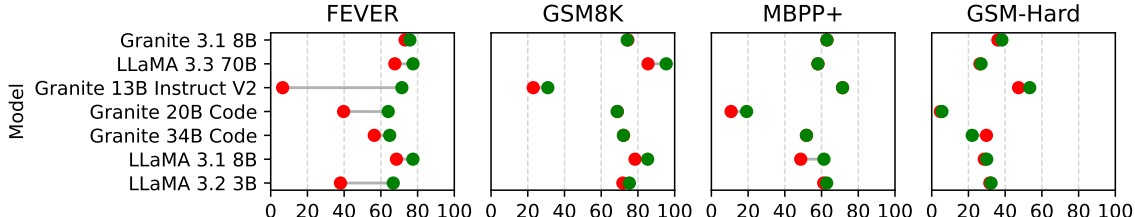

Figure A5: Comparison of optimized prompt program performance across models and datasets. Each barbell shows the accuracy improvement, if any, over the zero-shot baseline.

In Figure A5, we visualize the results from Table 1 and Table 2.

### 8.7 Accuracy *vs.* iterations

In Figure A6, we visualize the accuracy across candidates versus the iterations of the optimization process, including a 95% confidence interval depicting the spread in accuracy across candidates. The confidence interval is computed using mean and 1,000 bootstraps. As the iterations increase, the number of candidates decreases, while the size of the validation set $D_v$ increases.

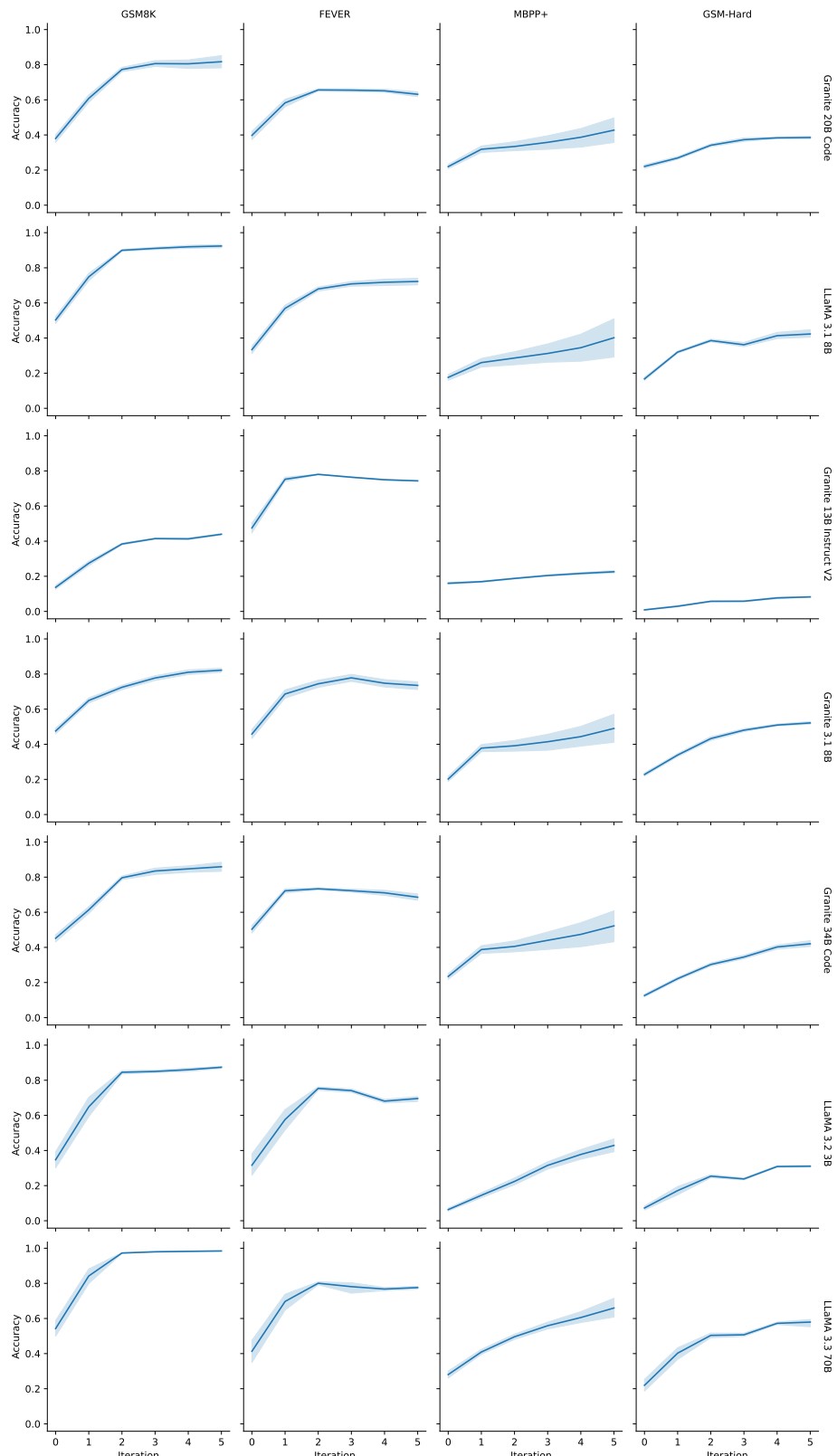

Figure A6: Accuracy *vs.* iterations with 95% confidence interval.

