# OpenReview forum: "AutoPDL: Automatic Prompt Optimization for LLM Agents"
_automl.cc/AutoML/2025/Methods_Track — AutoML 2025 Methods Track_

### Official Review · Reviewer_siAB · 2025-04-06

**Comments To Authors:**

This paper presents AutoPDL, an automated approach to optimize LLM prompts that is agnostic to the underlying task or LLM.
The problem is framed as an algorithm selection task which is solved using successive halving.

- The contribution is solid and technically sound.
- Well-written paper, clear and easy to follow
- Line 49: ⟨𝑛, 𝑑train, instr⟩ ∈ P is the prompt.  A concrete example would be beneficial for the reader
- As noted by the authors, additional approaches for optimization besides successive halving could be explored
- Missing a discussion on the implications (or necessity) of a larger exploration space

**Review Confidence:**

3

**Review Rating:**

8

---

### Official Review · Reviewer_dgCz · 2025-04-14

**Comments To Authors:**

**Summary**: Authors adopt a successive halving algorithm (genetic algorithm that keeps 1/2 of the “fittest” samples) to iteratively find the best prompting pattern and prompt combination for a given LLM. Experiments are conducted on various widely used datasets such as GSM8K, FEVER, and MBPP+ along with up to 70B scale models.

**Strengths**: A conceptually simple algorithm is shown to work quite well at finding good prompting patterns and prompts for smaller scale LLMs.

**Weaknesses**: Not much novelty in the approach and no analysis of the limitations.

**Detailed Feedback**:
- Line 45: Unclear what the difference between $D_{\mathrm{valid}}$ and $D_{\mathrm{test}}$ is
- Algorithm 1 (pg 13): don’t we need to include max validation subset size? Also specify how $D_v$ obtained. Is it sampled?
- Line 127: Should not assume readers know what trajectories $traj_i$ are. Should definitely what it is and also explain how it differs from the pairs $x_i, y_i$ supplied for CoT.
- Line 137: Algorithm 1 specifies reduction factor $\eta$ is variable but text says it’s fixed to $2$. Should be consistent.
- Section 5: It’s good to see a simple algorithm have solid results at ICL. However, there’s little innovation on the algorithmic side and no ablation studies on how the algorithm may have been improved. Maybe worth including some studies varying parameters of the algorithm (e.g introducing random mutations, changing the reduction ratio $\eta$)
- Before section 7, it would be good to have a section describing the limitations of this approach. For example, can successive halving without random mutations (i.e allowing some fraction of poorly performing candidates to survive) lead to a higher chance of landing at local optima?

**Recommendation**: Although some novelty in the work is missing, I believe authors still perform a solid empirical study of how a simple search algorithm can give boosts in LLM performance. Would recommend weak accept.

**Review Confidence:**

3

**Review Rating:**

7

---

### Official Review · Reviewer_Du8V · 2025-04-29

**Comments To Authors:**

This paper introduces AutoPDL, an offline optimization framework for automatically discovering optimal prompting strategies and prompt contents for large language models (LLMs). The authors formulate the prompt search space as a combination of prompting patterns (Zero-shot, Chain-of-Thought, ReAct, ReWOO) and textual prompt elements (few-shot examples and instructions), expressed using a human-readable, YAML-based prompting language called PDL. AutoPDL applies successive halving to explore this space efficiently. The approach is evaluated across three tasks (question answering, math, and programming) and six LLMs ranging from 8B to 70B parameters.

The paper is well organized and clearly written. The introduction motivates the problem well, and the methodology is explained in detail. However, the current scope is limited to a relatively narrow set of prompting patterns and pre-built templates, and the methodology needs to address broader generalization and robustness challenges.

Overall, the technical formulation is sound, and the offline optimization setup is correctly implemented. The use of successive halving is appropriate for efficient exploration given the combinatorial nature of the space. That said, several technical limitations reduce confidence in the completeness of the results:

Baseline Evaluation: The paper only compares AutoPDL's optimized prompts against a zero-shot baseline. It does not benchmark against alternative prompt optimization methods or search strategies (e.g., random search, Bayesian optimization, evolutionary methods). This weakens claims regarding the effectiveness of successive halving itself.

Agentic Trajectory Construction: The few-shot trajectories used for agentic prompting (ReAct, ReWOO) are synthetically generated via simple rule-based templates. While pragmatic, this reduces the ecological validity of the few-shot examples and may artificially favor the optimization. Additionally, most of the few-shot improvements could come from the format rather than the content of the examples.

Lack of Cross-Validation: The authors optimize over a single validation split without performing cross-validation or evaluating variability across multiple validation sets. Given the high variance in LLM performance on prompt configurations, this choice reduces the robustness of the reported gains.

Search Space: The optimization space is restricted to a small set of prompting patterns and a few discrete demonstration sizes (0, 3, or 5). Broader or more continuous search spaces are not explored. What about hyperparameters, such as temperature, for example? The configurations of the models are not reported in the paper.

Statistical Significance:
While the authors report average improvements in task performance after optimization, it does not provide any statistical significance analysis to support its claims. Given the high variance observed in LLM outputs across different prompts, it is essential to report in tables standard deviations and conduct hypothesis testing to ensure that the reported gains are not due to random chance. Without such statistical validation, the reliability of the performance improvements in tables remains uncertain.

**Review Confidence:**

4

**Review Rating:**

3

---

### Official Review · Reviewer_BF9i · 2025-04-30

**Comments To Authors:**

&nbsp;

**__SUMMARY__**

&nbsp;

The authors introduce AutoPDL, a prompt optimization approach that jointly considers the prompt pattern, instruction, and few-shot examples to be included within the prompt. Additionally, the authors make use of successive halving to improve optimization efficiency. The paper is well-written and the experiments have the potential to be informative for the community. As such, I recommend acceptance with the following points the authors may wish to consider.

&nbsp;


**__MAJOR POINTS__**

&nbsp;

1. For the results in Table 1 and Table 2 why did the authors not perform multiple random trials given the low runtimes?

2. In terms of the codebase, there appear to be a couple of files with commented out code. It would be great if the authors could perform a clean-up as well as e.g. running the code through an LLM (GPT/Claude) to document it.

3. In terms of related work it would be worth mentioning the Aviary library [3] which can also perform joint optimization over prompt pattern, instruction, and few-shot examples although it would require the definition of a custom op, as well as black-box prompt optimization methods [4-8].

4. The submission checklist appears to be missing from the submission.

&nbsp;

**__MINOR POINTS__**

&nbsp;

1. In the introduction, when mentioning different prompt patterns, it would also be worth mentioning zero-shot chain-of-thought [1].

2. It would be great if table captions could be included above rather than below the tables.

3. In Section 4.1, for the GSM8K dataset it would be great if the exact size could be given as there are some discrepancies between the sizes quoted e.g. in the original paper 8.5k questions are mentioned, however on HuggingFace there are 8.79k entries. From the authors' subsequent description I'm inferring that there are 8,449 questions in the full dataset?

4. Similarly in Section 4.1, what is the exact size of GSM-Hard (before splitting into D_valid and D_test) accounting for the removal of the 132 questions with incorrect ground truth answers?

5. What is the size of the dataset for MBPP+?

6. In Figure 7, how is the 95% confidence interval computed?

7. In Table 1, why is LLaMA 3.1 70B significantly worse than LLaMA 3.1 8B?

8. It would be worth updating reference [2] to the journal version of the paper (unpublished at time of submission to AutoML).

&nbsp;

**__REFERENCES__**

&nbsp;

[1] Kojima, T., Gu, S.S., Reid, M., Matsuo, Y. and Iwasawa, Y., 2022. [Large language models are zero-shot reasoners.](https://proceedings.neurips.cc/paper_files/paper/2022/hash/8bb0d291acd4acf06ef112099c16f326-Abstract-Conference.html) Advances in Neural Information Processing Systems, 35, pp.22199-22213.

[2] Yuksekgonul, M., Bianchi, F., Boen, J., Liu, S., Lu, P., Huang, Z., Guestrin, C. and Zou, J., 2025. [Optimizing generative AI by backpropagating language model feedback.](https://www.nature.com/articles/s41586-025-08661-4) Nature, 639(8055), pp.609-616.

[3] Narayanan, S., Braza, J.D., Griffiths, R.R., Ponnapati, M., Bou, A., Laurent, J., Kabeli, O., Wellawatte, G., Cox, S., Rodriques, S.G. and White, A.D., 2024. [Aviary: training language agents on challenging scientific tasks.](https://arxiv.org/abs/2412.21154) arXiv preprint arXiv:2412.21154.

[4] Cheng, J., Liu, X., Zheng, K., Ke, P., Wang, H., Dong, Y., Tang, J. and Huang, M., 2024, August. [Black-Box Prompt Optimization: Aligning Large Language Models without Model Training.](https://aclanthology.org/2024.acl-long.176/) In Proceedings of the 62nd Annual Meeting of the Association for Computational Linguistics (Volume 1: Long Papers) (pp. 3201-3219).

[5] Lin, X., Dai, Z., Verma, A., Ng, S.K., Jaillet, P. and Low, B.K.H., 2024. [Prompt optimization with human feedback.](https://arxiv.org/abs/2405.17346) arXiv preprint arXiv:2405.17346.

[6] Wu, Z., Lin, X., Dai, Z., Hu, W., Shu, Y., Ng, S.K., Jaillet, P. and Low, B.K.H., [Prompt Optimization with EASE? Efficient Ordering-aware Automated Selection of Exemplars.](https://openreview.net/forum?id=6uRrwWhZlM) In The Thirty-eighth Annual Conference on Neural Information Processing Systems.

[7] Hu, W., Shu, Y., Yu, Z., Wu, Z., Lin, X., Dai, Z., Ng, S.K. and Low, B.K.H., 2024. [Localized zeroth-order prompt optimization.](https://proceedings.neurips.cc/paper_files/paper/2024/hash/9cef1316eaef9bd99da46f63334dc031-Abstract-Conference.html) Advances in Neural Information Processing Systems, 37, pp.86309-86345.

[8] Zhou, Y., Muresanu, A.I., Han, Z., Paster, K., Pitis, S., Chan, H. and Ba, J., 2022, November. [Large language models are human-level prompt engineers.](https://openreview.net/forum?id=92gvk82DE-) In The Eleventh International Conference on Learning Representations.

&nbsp;

**Review Confidence:**

5

**Review Rating:**

8

---

### Meta-Review · Area_Chair_PCSs · 2025-05-11

**Recommendation:** Accept
**Confidence:** 4

**Metareview:**

This paper proposes a method called AutoPDL for prompt optimization that jointly optimizes the prompt pattern, instruction, and few-shot examples. Strengths of the paper include the simplicity of the proposed approach, its sound technical formulation, and informative set of experiments. The paper is well-written and organized, and appears to be reproducible as well.

In terms of points to improve, some concerns were raised in the reviews about the soundness of experimental validation, the novelty, and the breadth of exploration. Results were compared against a zero-shot baseline and not against other prompt optimization methods. Additionally, results were not collected over multiple trials which makes the statistical significance of the results unclear. There are also no ablation studies to indicate how the algorithm may be improved. Moreover, the approach has relatively low technical novelty and has a somewhat narrow scope in terms of patterns and templates.

There was some disagreement in reviewer opinion of this paper and this was a difficult decision. Although there are some remaining concerns about experimental validation, the proposed approach and corresponding experiments are likely to be of interest to the community. I recommend acceptance.